# Consecutive deletions in a unique Uruguayan SARS-CoV-2 lineage evidence the genetic variability potential of accessory genes

Yanina Panzera[1☯], Lucía Calleros[1☯], Natalia Goñi[2], Ana Marandino[1], Claudia Techera[1], Sofía Grecco[1], Natalia Ramos[3], Sandra Frabasile[3], Gonzalo Tomás[1], Emma Condon[1], María Noel Cortinas[2], Viviana Ramas[2], Leticia Coppola[2], Cecilia Sorhouet[4], Cristina Mogdasy[2], Héctor Chiparelli[2], Juan Arbiza[3], Adriana Delfraro[3]*, Ruben Pérez[1]*

1 Departamento de Biología Animal, Sección Genética Evolutiva, Instituto de Biología, Facultad de Ciencias, Universidad de la República, Montevideo, Uruguay, 2 Departamento de Laboratorios de Salud Pública, Centro Nacional de Referencia de Influenza y otros Virus Respiratorios, Ministerio de Salud Pública, Montevideo, Uruguay, 3 Facultad de Ciencias, Sección Virología, Instituto de Biología e Instituto de Química Biológica, Universidad de la República, Montevideo, Uruguay, 4 Laboratorio de Biología Molecular, Mutualista Médica Uruguaya, Montevideo, Uruguay

☯ These authors contributed equally to this work.
* adelfraro@fcien.edu.uy (AD); rperez@fcien.edu.uy (RP)

**Data Availability Statement:** Yes - all data are fully available without restriction Viral sequences were deposited in the GenBank with the accession

## Abstract

Deletions frequently occur in the six accessory genes of SARS-CoV-2, but most genomes with deletions are sporadic and have limited spreading capability. Here, we analyze deletions in the ORF7a of the N.7 lineage, a unique Uruguayan clade from the Brazilian B.1.1.33 lineage. Thirteen samples collected during the early SARS-CoV-2 wave in Uruguay had deletions in the ORF7a. Complete genomes were obtained by Illumina next-generation sequencing, and deletions were confirmed by Sanger sequencing and capillary electrophoresis. The N.7 lineage includes several individuals with a 12-nucleotide deletion that removes four amino acids of the ORF7a. Notably, four individuals underwent an additional 68-nucleotide novel deletion that locates 44 nucleotides downstream in the terminal region of the same ORF7a. The simultaneous occurrence of the 12 and 68-nucleotide deletions fuses the ORF7a and ORF7b, two contiguous accessory genes that encode transmembrane proteins with immune-modulation activity. The fused ORF retains the signal peptide and the complete Ig-like fold of the 7a protein and the transmembrane domain of the 7b protein, suggesting that the fused protein plays similar functions to original proteins in a single format. Our findings evidence the remarkable dynamics of SARS-CoV-2 and the possibility that single and consecutive deletions occur in accessory genes and promote changes in the genomic organization that help the virus explore genetic variations and select for new, higher fit changes.

numbers: MZ555811 to MZ555814 and OK416091 to OK416099.

**Funding:** This work was supported by the Facultad de Ciencias and Comisión Sectorial de Investigación Científica (CSIC) (Grant CSIC Equipamiento) Plataforma Genómica Facultad de Ciencias. Responsable YP. Fundación Manuel Pérez, UdelaR (Grant Fondo Manuel Pérez). Responsable RP, JA and CM. The funders had no role in study design, data collection and analysis, decision to publish, or preparation of the manuscript.

**Competing interests:** The authors have declared that no competing interests exist.

# Introduction

Severe acute respiratory syndrome coronavirus-2 (SARS-CoV-2) is a novel Betacoronavirus of the subgenus *Sarbecovirus* and the causative agent of the pandemic coronavirus disease 2019 (COVID-19) [1].

The SARS-CoV-2 genome is a ~ 30 kb long, single-stranded, positive RNA molecule, with the typical gene organization of coronaviruses. There are 12 open reading frames (ORFs) that encode 26 proteins, including 16 non-structural proteins (nsp1 to nsp16), four structural proteins (M, N, S, and E), and six accessory proteins (3a, 6, 7a, 7b, 8, 10). Accessory proteins are dispensable for replication in cell culture but may have regulatory roles during the viral cycle and thus contribute to the virus fitness by increasing the ability to evade the host's innate immune response [2,3]. Coronavirus groups usually differ in those accessory proteins, and more infective species have specific pathogenic sets [4].

SARS-CoV-2 has accumulated many variations since its emergence in late 2019, including some mutations in the viral spike (S) protein that underwent rapid growth and spread. These S changes have generated concern and interest, particularly the D614G, N501Y, E484K, and K417N amino acid replacements that have occurred in the receptor-binding domain [5].

Nucleotide substitutions that produce amino acid replacements constitute the primary raw material for genetic variation; however, deletions and insertions (indels) are critical elements in the coronavirus macro and microevolution [6,7]. There is also an apparent link between both processes, as some punctual mutations occur near indels [8].

Indels in coronaviruses remain uncorrected by the proofreading activity of nsp14-exoribonuclease and enhanced by the discontinuous RNA synthesis of the polymerase machinery [5]. Although most indels likely negatively affect viral fitness [9], a small number emerge and spread in viral populations, suggesting a positive effect on viral fitness and adaptive evolution. Thus, the analysis of these indels may reveal evolutionary trends and provide new insight into the surprising variability and rapidly spreading capability that SARS-CoV-2 has shown since its emergence.

Recent evidence establishes the existence of recurrent deletion regions that map to defined antibody epitopes [10]. Furthermore, these deletions appear to emerge independently in a convergent pattern of viral antigenic evolution that may confer resistance to neutralizing antibodies. An excellent example of these recurrent deletions is those acquired in the N-terminal domain of the S glycoprotein during long-term infections of immunocompromised patients, supporting a role in the evolution by altering defined antibody epitopes [10,11].

Deletions also occur frequently in accessory ORFs with interesting outcomes and potential effects on virus evolution [2,12]. These ORFs modulate the host's immune response and are becoming the focus of active research to decipher their role in viral pathogenicity. The ORF7a and ORF7b are two contiguous genes, with a short overlap of four nucleotides, which undergo deletions that could strongly affect protein structure [13]. Studies performed in the homologous ORF7a and 7b of SARS-CoV-1 have provided insights into their structure and functions. Both ORFs encode transmembrane proteins localized in the endoplasmic reticulum and Golgi network [14].

The 121-residue protein 7a (p7a) of SARS-CoV-2 consists of a 15-residue N-terminal signal peptide, an 81-residue luminal domain, a 20-residue transmembrane domain (TMD), and a 5-residue cytosolic tail [15]. The luminal domain has a 7-stranded β-sandwich fold typical of the immunoglobulin (Ig) superfamily [16].

The 43-residue protein 7b (p7b) of SARS-CoV-2 has 85.4% identity with its SARS-CoV-1 homologous [15] and consists of an 8-residue N-terminus (external), a 22-residue TMD, and a 13-residue C-terminal domain [14].

Previously, we detected a 12-nucleotide deletion (Δ12) in ORF7a that emerged in a single outbreak of a particular B.1.1.33 clade that was later renamed N.7 lineage [17]. Here, we show that the Δ12 variant is widely extended in the Uruguayan unique N.7 lineage and identified a new variant with two deletions, the already reported Δ12 and a subsequent downstream novel 68-nucleotide deletion in the same ORF7a that produce a fused ORF7ab. Our findings evidence the remarkable variation dynamics of SARS-CoV-2 and the possibility that consecutive deletions occur in accessory genes to promote changes in genomic organization and structure.

## Materials and methods

### Samples and SARS-CoV-2 diagnosis

Combined nasopharyngeal and oropharyngeal swab samples from Uruguayan patients were collected from July to October 2020 by the Reference Center for Influenza and other Respiratory Viruses, National Institute of Health Laboratories (DLSP-MSP). The collection and analysis of samples were performed according to the Declaration of Helsinki; no specific authorization was required because the activities were conducted as part of routine virological surveillance (anonymously, without identification of patients) by the Uruguayan official Institution (DLSP-MSP). The SARS-CoV-2 diagnosis was performed by RNA extraction with the Qiamp Viral RNA Minikit (Qiagen USA) followed by real-time reverse transcription-polymerase chain reaction (RT-qPCR) using the protocol recommended by the Panamerican Health Organization (PAHO-WHO) [18].

### PCR, Sanger sequencing, and capillary electrophoresis

An aliquot of RNA (12μL) was used for cDNA synthesis with Superscript II ® reverse transcriptase (Thermo Fisher, USA) and random primers. cDNA was analyzed by PCR with a single pair of primers to screen for the Δ12 variant by amplifying an ORF7a fragment of approximately 406 bp [17]. Amplicons were analyzed by capillary electrophoresis on a Fragment Analyzer 5200 system using the High Sensitivity NGS Analysis Kit (Agilent Technologies, USA). Amplicons were also subjected to Sanger sequencing in Macrogen (Korea).

### Full-length genome amplification using ARTIC protocol

The SARS-CoV-2 genome of thirteen samples was enriched using ARTIC primer scheme version 3; information on the primer sequences and protocols are available at the ARTIC network repository (https://artic.network/ncov-2019).

The Nextera DNA Flex Library Preparation kit (Illumina, USA) with dual indexing was used for library preparation. The libraries were purified with AMPure XP beads (Beckman Coulter, USA). All quantifications were carried out using a Qubit DNA High-Sensitivity kit (Invitrogen, USA). The library's control quality and length were assessed on a Fragment Analyzer 5200 system (Agilent Technologies, USA) using the Standard Sensitivity NGS Analysis Kit (Agilent Technologies, USA).

Whole-genome sequencing was performed on an Illumina MiniSeq (Illumina, USA) platform using MiniSeq™ Mid Output Reagent Cartridge (300-cycles, paired-end reads). Adapter/quality trimming and filtering raw data were performed with BBDuk, and clean reads were mapped to the consensus genome using Geneious Prime 2020.1.2 (https://www.geneious.com).

### Deletion characterization and comparison

The online web application CoV-GLUE (http://cov-glue.cvr.gla.ac.uk/#/deletion) was used to assess genetic changes (single nucleotide polymorphisms and indels) in SARS-CoV-2 genomes [19].

### SARS-CoV-2 lineage assignment and phylogenetic analysis

The lineage of the strains was assigned according to the nomenclature system proposed by Rambaut et al. [20] using the Pangolin tool [21].

For phylogenetic analysis, all Uruguayan full-length sequences in the GISAD EpiCoV database [22] were retrieved with associated metadata to generate a comprehensive genome dataset of around eight hundred sequences. The dataset was filtered and reduced to over five hundred for complete sequences with good coverage and quality for further comparative analysis.

DNA alignments were performed with MAFFT [23]. With 1000-replicates bootstrap to support internal nodes, maximum-likelihood trees were inferred in Geneious using FastTree [24] and visualized with the ggTree package in R.

### Protein structure

The structure of the p7a was obtained from the protein database (PDB ID: 6W37) and I-TAS-SER server (https://zhanglab.ccmb.med.umich.edu/COVID-19/).

Transmembrane helices were predicted using a Transmembrane Hidden Markov model (TMHMM) available as a Geneious plugin.

## Results

### Detection of deletions and genome sequencing

We screened 51 SARS-CoV-2 samples for the presence of the ORF7a deletion (Δ12) in Uruguay. The screening was expected to produce two bioanalyzer electropherogram patterns corresponding to a 406-nt amplicon (wild type variant) and a 394-nt amplicon (Δ12 variant). Nine samples had the Δ12 deletion patterns (394-nt amplicon), but four samples showed a pattern corresponding to an amplicon of approximately 326 nucleotides, indicative of putative larger deletion/s (Fig 1A and 1B). The results of Sanger sequencing evidenced that these smaller amplicons had two deletions, the previous Δ12 and a novel 68-nucleotide deletion (Δ68), located 44 nucleotides downstream in the ORF7a. All four individuals had the same two deletions (12 and 68 nucleotides) confirmed by Sanger sequencing (Fig 1C).

We obtained and analyzed the thirteen genomes with single and double deletions. The complete genomes had good average coverage (>1000×) and quality (none or very few unassigned bases). Comparison with the Wuhan-Hu-1 reference showed conserved amino acid substitutions in the following positions: L37F (ORF1ab, nsp6), P323L (ORF1ab, nsp12), D614G (S), I33T (ORF6), and four changes in N (P168S, R203K, G204R, I292T). Previous sequences with the 12-nt deletion obtained from Uruguay (Genbank Accession numbers: MW298637-MW298643) have the same residue differences and almost identical sequences.

All genomes belonged to the N.7 lineage of SARS-CoV-2 (also called B.1.1.33.7). Genomes came from patients that tested positive for SARS- CoV-2 and had different clinical manifestations, including asymptomatic or mild infections with fever, headache, asthenia, cough, ageusia, and odynophagia (Table 1). Most patients recovered without treatment, but two with preexisting conditions (cerebrovascular accident or cancer) died after intensive care treatment (Table 1).

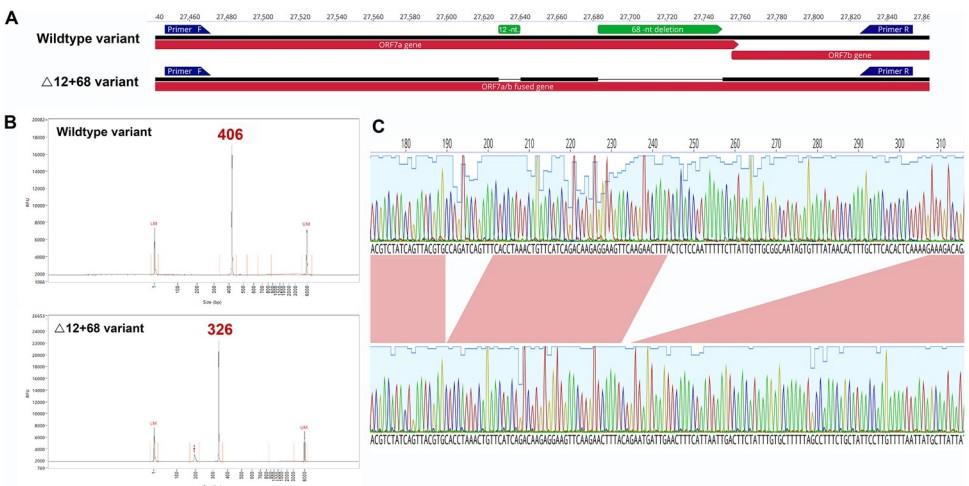

**Fig 1.** A: ORF7a amplicon for the wildtype and Δ12+68 variants. Artic primers annotations are visualized by blue arrows. B: Chromatogram peaks after capillary electrophoresis, wildtype, and Δ12+68 variants. C: Zoomed diagram details nucleotide sequence of the double deletion obtained by Sanger sequencing.

Samples came from different Uruguayan departments (Montevideo, Soriano, Canelones, Florida, and Maldonado) and were collected from July to October 2020. Some patients were epidemiological related, particularly two with the double deletion variant that live in the same house (Table 1).

## Phylogenetic analysis

The newly sequenced genomes from the N.7 lineage were added to the Uruguayan genomes retrieved from the GISAID (n = 497). This dataset included all the available good-quality N.7 sequences in the databases (n = 40). No additional N.7 sequences were available from other countries, including bordering Argentine and Brazil.

The N.7 lineage samples (n = 40) appeared in a single branch in the phylogenetic tree, agreeing with its lineage classification. Sequences of the N.7 lineage with deletions (Δ12 and

**Table 1. Epidemiological data of the samples analyzed in the study.**

| Sample | Accession | Date | Origin | Sex | Age | Clinical data | Epidemiological data |
|--------|-----------|------|--------|-----|-----|---------------|----------------------|
| 299 | OK416091 | 7/16/2020 | Canelones | M | 67 | preexisting condition (cerebrovascular accident), died | Shared ICU with 300 |
| 300 | OK416092 | 7/16/2020 | Canelones | F | 74 | preexisting condition (cancer), died | Shared ICU with 299 |
| 649 | OK416093 | 8/19/2020 | Montevideo | M | 69 | symptomatic mild infection/recovered | NA |
| 731 | OK416094 | 9/1/2020 | Florida | M | 73 | asymptomatic | NA |
| 799 | OK416095 | 9/9/2020 | Canelones | F | 63 | symptomatic mild infection/recovered | NA |
| 822 | OK416096 | 9/14/2020 | Montevideo | F | 7 | asymptomatic | NA |
| 839* | MZ555813.1 | 9/16/2020 | Montevideo | M | 64 | symptomatic mild infection/recovered | NA |
| 879* | MZ555814.1 | 9/17/2020 | Montevideo | M | 46 | asymptomatic/recovered | NA |
| 840 | OK416097 | 9/16/2020 | Maldonado | M | 65 | asymptomatic/recovered | NA |
| 899* | MZ555811.1 | 9/22/2020 | Canelones | F | 49 | asymptomatic/recovered | household contact with 917 |
| 917* | MZ555812.1 | 9/22/2020 | Canelones | M | 52 | asymptomatic/recovered/reinfected | household contact with 899 |
| 1021 | OK416098 | 10/5/2020 | Soriano | F | 34 | symptomatic/recovered | NA |
| 1102 | OK416099 | 10/16/2020 | Montevideo | M | NA | symptomatic/recovered | NA |

All samples have the 12-nt deletion; samples with asterisks also have the 68-nt deletion. NA: Not available data.

Δ12+Δ68) fell in a separated branch from samples without deletions collected in the Uruguayan department of Treinta y Tres during June 2020 (Fig 2). This branch of N.7 genomes carrying deletions (n = 20) showed amino acid replacements in nsp6 (L37F) and N (R36Q).

The four genomes with the double deletion (Δ12+Δ68) were identical in the coding and non-coding sequences and had a characteristic amino acid replacement in nsp3 (T976I) lacking in other N.7 sequences.

## Deletion effects: The emergence of a fused ORF

The known Δ12 (position: 27629–27640) and the novel frameshifting Δ68 (position: 27683–27750) occur at the end of the ORF7a (27394–27759) (Fig 3, top). The Δ12 results in a 4-aa deletion (RSVS) in position 80–83 of the 121-aa protein 7a. The Δ68 produces a 23-aa deletion after position 96 and a frameshifting that alters the stop codon and creates an in-frame fusion with the complete ORF7b (44 codons). The simultaneous occurrence of both deletions (Δ12 +Δ68) joins the deleted ORF7a and ORF7b, leading to a new ORF7ab with 138 codons, including the ORF7b's TAA stop codon (Fig 3, top). The first 94 residues of the putative protein ORF7ab (p7ab) belong to the p7a amino end and comprise the signal peptide and all the seven

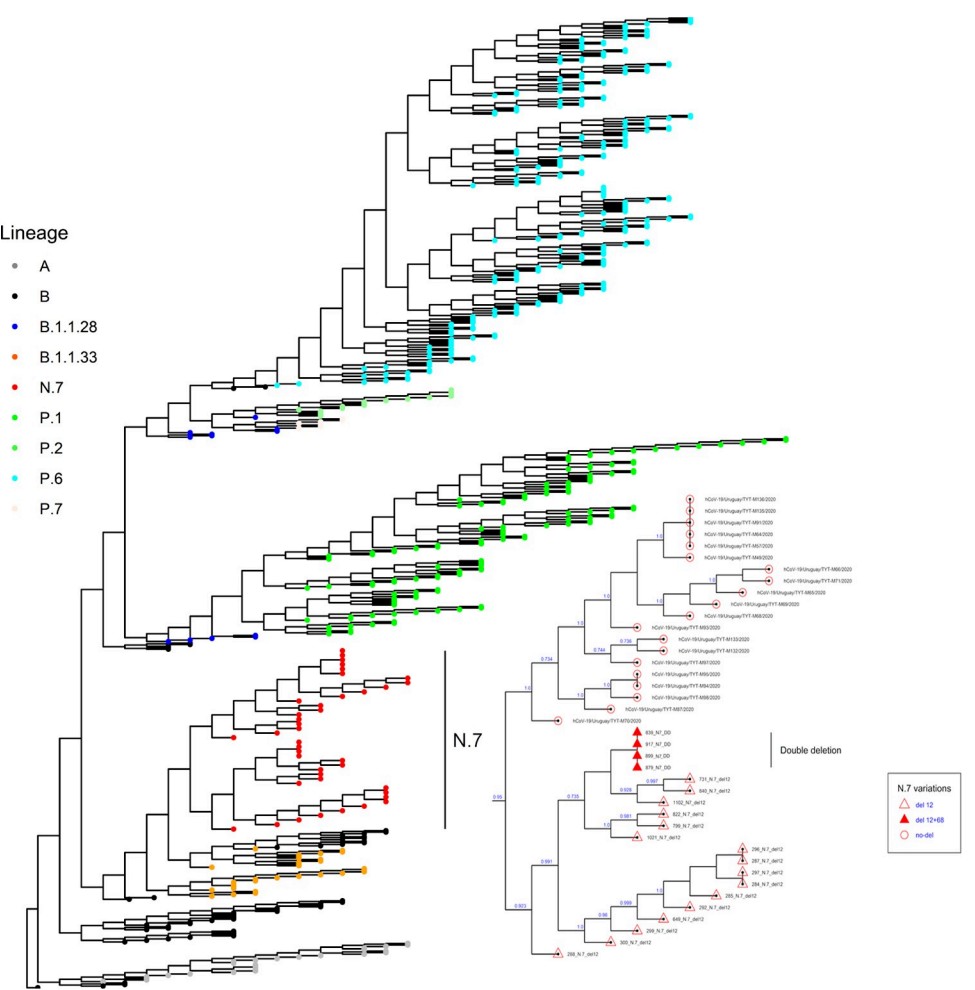

**Fig 2. Maximum-likelihood tree based on 497 SARS-CoV-2 sequences.** Details of the N.7 lineage composition are zoomed on the right.

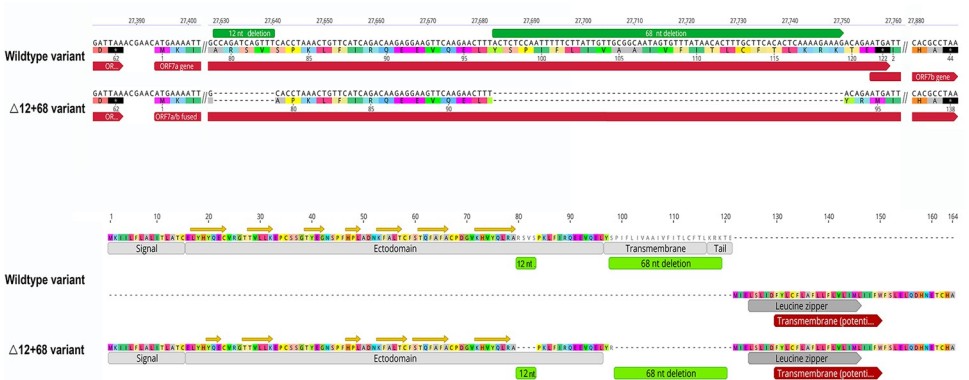

**Fig 3.** Top: Nucleotide and amino acid sequences of the ORF 7a/7b in wildtype and Δ12+68 variants. Green bars show the deletion regions. Bottom: Primary structure of the 137-aa 7ab putative protein is compared with the 121-aa 7a protein of the wildtype variant. The different domains are annotated. Yellow arrows denote ß-strands.

β-strands of the ectodomain Ig-fold; deletions remove most of the 20-residue TMD and the entire 5-residue cytoplasmic tail.

The primary sequence of the p7b, consisting of both terminal regions and the central TMD is wholly embedded in the fused p7ab. Therefore, the fused putative protein has only one TMD from the p7b. Transmembrane domains were annotated based on reported data and predicted *de novo* for the ORF7a, ORF7b, and fused ORF7ab (Fig 3, bottom). The predicted TMD for both proteins has the same size (21 residues) and co-localizes with the reported TMD of 20 (p7a) and 22 residues (p7b).

## Discussion

Deletions have played a significant role in the evolution of the *Coronaviridae* family. The large coronavirus RNA genome is prone to deletions and less affected by the loss of genetic material than other viruses. Among the possible reasons for this feature is the existence of accessory genes that exhibit high plasticity and resilience to undergo indel changes [6]. Remarkably, certain deletions could be so extensive that they eliminate entire accessory ORFs. For example, in Europe and Asia, two phylogenetically unrelated deletions completely removed the 7a, 7b, and 8 consecutive accessory ORFs [25,26]. These extreme deletions pose an interesting question about the role of accessory ORFs in virus viability and reinforce the relevance of deletions in increasing SARS-CoV-2 variability during expansion in the human population.

Some deletions in accessory genes may be related to coronavirus adaptation to the host cell environment. In SARS-CoV-1, a 45-nucleotide deletion in the ORF7b emerged following passage in Vero E6 cells, suggesting attenuation [27]. The same deletion occurs in other cultured strains [28,29]. A 382-nt deletion in ORF8 is associated with milder SARS-CoV-2 infection [30]. Three different deletions in ORF7a truncate the C-terminal half of the protein; two of them result in growth defects and failures to suppress the immune response, providing potential advantages toward the SARS-CoV-2 adaptation to humans [6,31–33].

Here we detected single (Δ12) and double deletion (Δ12+68) variants of SARS-CoV-2 within the N.7 lineage (Fig 2). This lineage derived from the Brazilian B.1.1.33 lineage predominant in Brazil during the initial part of the pandemic [31]. In Uruguay, the N. 7 lineage emerged in June 2020 and was replaced during November and December 2020 by Gamma (P.1) and Zeta (P.2) strains derived from the B.1.1.28 lineage [32].

The phylogenetic relatedness (Fig 2), the timing of collection, and the local geographic distribution support that the N.7 lineage emerged in Uruguay during the early pandemic and

likely emerged from the B.1.1.33 lineage that lacks the Δ12 variant. Furthermore, the ancestral N.7 lineage also lacked the Δ12 change, as supported by undeleted strains collected during June 2020 in the Treinta y Tres department bordering Brazil.

The topology of the phylogenetic tree implies that the Δ12 variant emerged early in the N.7 lineage, during its initial differentiation in Uruguay (the first case was reported in July 2020), and that the Δ12+68 variant emerged subsequently by the acquisition of a second deletion (Δ68) from a Δ12 individual. To our knowledge, this is the first case where the timing of occurrence of two consecutive deletions in an accessory gene is identified in SARS-CoV-2. It is relatively uncommon to reconstruct the evolutionary pathway of deletions. Indels in other regions, for instance, near the furin cleavage site in the S gene, frequently occur during coronavirus evolution [34]. However, the evolutionary path of these deletions is complex and difficult to determine by analyses of the underlying nucleotide sequence [8].

The Δ12+68 variant has the following characteristics making it unique for SARS-CoV-2: i) both deletions are restricted to a small region of the accessory ORF7a, ii) the successive occurrence of deletions creates a new ORF that retains key functional motifs of 7a and 7b proteins, iii) strains with both deletions are transmitted between individuals, and iv) individual deletions (Δ12 and Δ68) occur consecutively with a known order (the Δ12 occurred first while the Δ68 has a more recent origin).

Addetia et al. described two large and phylogenetically unrelated deletions (392 and 227 nucleotides long) that fuse accessory ORF7a with downstream ORFs [13]. The most extensive 392-nucleotide deletion creates a new ORF by connecting the N-terminus of ORF7a with ORF8. The 227-nucleotide deletion results in a new ORF by fusing the N-terminus of ORF7a with the entire ORF7b. The retained N-terminus of ORF7a in both deletions comprises only the signal peptide and two or three β-strands of the seven that form the β-sheets of the Ig-fold of ORF7a and lost the entire protein's transmembrane and cytoplasmic domains. The Ig-fold in the luminal domain of the p7a is the putative functional interface, suggesting that variability in this region affects binding capability and is largely deleterious for the virus.

The Δ12+68 variant here described is the first having a fused ORF that keeps the N-terminus signal and Ig-like domain of p7a and the TMD of p7b (Fig 3, bottom). In vitro studies showed that the p7a ectodomain attaches to human monocytes, decreasing antigen-presenting ability and inducing overexpression of pro-inflammatory cytokines [35]. The TMD and cytoplasmic tail mediate the intracellular retention within the Golgi network, suggesting a potential role in viral assembly or budding events in the endoplasmic reticulum to Golgi intermediate compartment (ERGIC) [16,31]. The entire p7b, included in the fused p7ab, provides a TMD that may localize the protein in the Golgi compartment [14,36]. This TMD is supposed to assemble into stable multimers through a leucine zipper and interfere with cellular processes that involve the leucine-zipper association [37]. Functional studies would be needed to establish if the fused protein functions similarly to p7a and p7b in a single format.

A limitation of the study is the low number of samples due to Uruguay's low prevalence that underpowers statistical analysis (i.e., frequency of cases and clinical outcome). Our analysis, therefore, represents a snapshot in time before the emergence of variants of concern that replaced the N.7 lineage. Sampling was also affected by the relatively reduced number of sampling sites (Table 1); individuals submitted for the report may not fully reflect the prevalence of infections in the overall population. Despite the above limitations, the deletion frequency in the sample (~25%) is significant in the context of the scenario in Uruguay during 2020, characterized by very few outbreaks and cases with epidemiological connections [38].

Although deletions frequently occur in accessory ORFs, the occurrence of two consecutive deletions suggests a functional synergy in viral infections and constitutes evidence that consecutive deletions also act in SARS-CoV-2 accessory genes. Fitness analysis of both deleted

variants would be needed to establish whether the second deletion was neutral or had a selective advantage (i.e., increased transmission or milder symptoms) during certain stages of the SARS-CoV-2 infection. We examined the relationship between deletions and the patient's symptomatology. Patients with deletions exhibit many clinical signs, and two patients with the single deletion (Δ12) died after treatment. All patients with double deletions recovered from the infection, three were asymptomatic, and one had only mild symptoms (Table 1). Although this association suggests a lesser clinical effect of deleted viruses, the results are inconclusive and should be corroborated in global studies with deletions occurring in different SARS-CoV-2 lineages.

Based on previous studies, some degree of indels variability was expected in coronavirus infecting human populations. Still, the current global scenario shows that the deletion potential in SARS-CoV-2 was initially sub-estimated and underdiagnosed before it gained full attention by the scientific community [7,13,25,26,39,40]. In this sense, our study constitutes an example of the impact of deletion in SARS-CoV-2 evolution and underscores the importance of indel variation in providing new insights into the genome variability and dynamics of accessory ORFs within SARS-CoV-2 lineages.

## Supporting information

**S1 Table. Acknowledgments table for sequences obtained from the GISAID's EpiCoV™ database.**
(PDF)

## Acknowledgments

We want to thank all the authors who have kindly deposited and shared genome data on GISAID (S1 Table).

## Author Contributions

**Conceptualization:** Yanina Panzera, Lucía Calleros, Ruben Pérez.

**Data curation:** Ruben Pérez.

**Formal analysis:** Yanina Panzera, Ruben Pérez.

**Funding acquisition:** Yanina Panzera, Cristina Mogdasy, Juan Arbiza, Ruben Pérez.

**Investigation:** Yanina Panzera, Lucía Calleros, Natalia Goñi, Ana Marandino, Claudia Techera, Sofía Grecco, Natalia Ramos, Sandra Frabasile, Gonzalo Tomás, Emma Condon, María Noel Cortinas, Viviana Ramas, Leticia Coppola, Cecilia Sorhouet, Adriana Delfraro, Ruben Pérez.

**Methodology:** Yanina Panzera, Lucía Calleros, Ruben Pérez.

**Project administration:** Ruben Pérez.

**Resources:** Ruben Pérez.

**Software:** Ruben Pérez.

**Supervision:** Yanina Panzera, Lucía Calleros, María Noel Cortinas, Cristina Mogdasy, Héctor Chiparelli, Adriana Delfraro, Ruben Pérez.

**Validation:** Ruben Pérez.

**Visualization:** Ruben Pérez.

**Writing – original draft:** Yanina Panzera, Lucía Calleros, Ruben Pérez.

**Writing – review & editing:** Yanina Panzera, Lucía Calleros, Natalia Goñi, Ana Marandino, Claudia Techera, Sofía Grecco, Natalia Ramos, Sandra Frabasile, Gonzalo Tomás, Emma Condon, María Noel Cortinas, Viviana Ramas, Leticia Coppola, Cecilia Sorhouet, Cristina Mogdasy, Héctor Chiparelli, Juan Arbiza, Adriana Delfraro, Ruben Pérez.

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
