## [Decision Letter · Decision Letter 0]

7 Jan 2022

PONE-D-21-32425Consecutive deletions in a unique Uruguayan SARS-CoV-2 lineage evidence the genetic variability potential of accessory genesPLOS ONE

Dear Dr. Perez,

Thank you for submitting your manuscript to PLOS ONE. After careful consideration, we feel that it has merit but does not fully meet PLOS ONE’s publication criteria as it currently stands. Therefore, we invite you to submit a revised version of the manuscript that addresses the points raised during the review process.

 As pointed out by the reviewer, while the technical analysis and writing are done very well, there is an overabundance of speculative discussion in the manuscript. It would take much more data and analysis to untangle the combined effect of viral fitness, population genetics, and public health measures on the spread and prevalence of particular SARS-CoV-2 lineages in this pandemic. We appreciate that these hypotheses were presented sensibly, but politely request that the discussions from line 280 onwards be scaled back to strike a more balanced tone. Please also address the other points raised by the reviewer below.

We look forward to receiving your revised manuscript.

Kind regards,

Herman Tse

Academic Editor

PLOS ONE

Journal Requirements:

Reviewers' comments:

Reviewer's Responses to Questions

**Comments to the Author**

1. Is the manuscript technically sound, and do the data support the conclusions?

Reviewer #1: Partly

2. Has the statistical analysis been performed appropriately and rigorously? 

Reviewer #1: N/A

3. Have the authors made all data underlying the findings in their manuscript fully available?

Reviewer #1: Yes

4. Is the manuscript presented in an intelligible fashion and written in standard English?

Reviewer #1: Yes

5. Review Comments to the Author

Reviewer #1: In this manuscript, the authors present evidence of circulation of strains with 2 deletions within the ORF7a gene in 2020. They speculate that the functional consequence of these deletions is the expression of a putative ORF7ab gene product. ORF7 deletions in SARS-CoV-2 (even to the extent of complete absence) have been reported from various parts of the world. The significance of such deletions on SARS-CoV-2 transmissibility or clinical manifestations is not known. The methodology of this study is clear and the results are convincing. The epidemiological and clinical significance of these ORF7-del variants is uncertain. But, I think there is considerable speculation in parts of the manuscript that needs to be revised.

Specific comments below:

- Please cite other papers describing ORF7b deletions: Tse H et al, Emergence of a Severe Acute Respiratory Syndrome Coronavirus 2 virus variant with novel genomic architecture in Hong Kong, Clin Infect Dis; Mazur-Panasiuk N et al, Expansion of a SARS-CoV-2 Delta variant with an 872 nt deletion encompassing ORF7a, ORF7b and ORF8, Poland, July to August 2021, Euro Surveillance.

- How many samples were analyzed in total to obtain the few with 12-nt deletion in table 1? Unable to gauge how common this variant is.

- Any epidemiological link between patients listed in table 1?

- Were any of the patients listed in table 1 immunocompromised?

- "This potential functionality could explain why the sizeable Δ12+68 change could be transmitted at least between four individuals." this seems to imply that these four patients transmitted this variant between each other, which I don't think is the case? Please clarify.

- The delta-68 deletion could also have occurred within 4 hosts independently (intra-host evolution) without being transmitted? So, if there is no epidemiological link between these 4 cases, the authors should not speculate on the transmissibility of this variant.

- "The topology of the phylogenetic tree implies that the Δ12 variant emerged early in the N.7 lineage during its initial differentiation in Uruguay (the first case was reported in July 2020); the Δ12+68 variant emerged subsequently by the acquisition of a second deletion (Δ68) from a Δ12 individual." I agree this is probably most likely, but any possibility of under-sequenced communities or provinces where an earlier emergence of the Δ68 might have been missed?

- I find the discussion on structure and function of the putative ORF7ab gene product to be very speculative (line 280 onwards). Should be edited and condensed heavily to avoid speculations (such as "the Δ12 mutation in ORF7a could be slightly deleterious and temporarily tolerated in a low-competitive population; after acquiring the additional deletion (Δ68)"). Instead, just state the type of experiments that could be performed to determine effects of deletions observed.

- Should add a section on limitations of the study.

6. PLOS authors have the option to publish the peer review history of their article (what does this mean?). If published, this will include your full peer review and any attached files.

Reviewer #1: No

---

## [Author Response · Author response to Decision Letter 0]

20 Jan 2022

The following comments were uploaded as "Response to Reviewers" file. 

Reviewer #1: In this manuscript, the authors present evidence of circulation of strains with 2 deletions within the ORF7a gene in 2020. They speculate that the functional consequence of these deletions is the expression of a putative ORF7ab gene product. ORF7 deletions in SARS-CoV-2 (even to the extent of complete absence) have been reported from various parts of the world. The significance of such deletions on SARS-CoV-2 transmissibility or clinical manifestations is not known. The methodology of this study is clear and the results are convincing. The epidemiological and clinical significance of these ORF7-del variants is uncertain. But, I think there is considerable speculation in parts of the manuscript that needs to be revised.

Specific comments below:

Please cite other papers describing ORF7b deletions: Tse H et al, Emergence of a Severe Acute Respiratory Syndrome Coronavirus 2 virus variant with novel genomic architecture in Hong Kong, Clin Infect Dis; Mazur-Panasiuk N et al, Expansion of a SARS-CoV-2 Delta variant with an 872 nt deletion encompassing ORF7a, ORF7b and ORF8, Poland, July to August 2021, Euro Surveillance.

Answer: We thank the reviewer for pointing out these two important papers; both references were added to the manuscript discussion. 

- How many samples were analyzed in total to obtain the few with 12-nt deletion in table 1? Unable to gauge how common this variant is.

Answer: We screened 51 samples that were available in our Uruguayan biobank of positive samples (this information was added to the manuscript). Therefore, the frequency of the double deletion is 13/51 (approximately 25%). However, we recognize that this number could be biased because we do not have access to all the samples from Uruguay. Therefore, we added a sentence about frequency in the manuscript and clarified this possible bias in the "limitation of the study" statement. 

- Any epidemiological link between patients listed in table 1?

Answer: In the study period, the epidemic in Uruguay was characterized by different outbreaks, with an average of about 20 new COVID-19 cases per day. Because of the reduced number, sixty-seven percent of patients had a contact record with a confirmed case in the national territory, and thirteen percent had known contact with a traveler (https://www.gub.uy/ministerio-salud-publica/comunicacion/noticias/informe-epidemiologico-covid-19-del-30-noviembre-2020). The genetic homogeneity of the samples also supports a few transmission clusters. 

We obtained and added new data to Table 1, evidencing the epidemiological link among some patients. Regretfully, this epidemiological data were not available for all patients. Therefore, we added a sentence clarifying this point and included the bias in the limitation of the study section. 

- Were any of the patients listed in table 1 immunocompromised?

Answer: We could obtain only detailed clinical information of the two patients that required admission to the intensive care unit. Both have preexisting conditions, one had a cerebrovascular stroke, and the other had cancer. Unfortunately, these patients died in the health center. These data were included in Table 1 and the manuscript's text. The data of the other patients did not indicate evidence of immunosuppression. 

- "This potential functionality could explain why the sizeable Δ12+68 change could be transmitted at least between four individuals." this seems to imply that these four patients transmitted this variant between each other, which I don't think is the case? Please clarify.

Answer: Two of the four Δ12+68 individuals are from the same family (new data added to Table 1). The four individuals were likely part of one transmission cluster, but the referee is right that we cannot establish the exact connection among them. Therefore, we removed this sentence from the discussion. 

- The delta-68 deletion could also have occurred within 4 hosts independently (intra-host evolution) without being transmitted? So, if there is no epidemiological link between these 4 cases, the authors should not speculate on the transmissibility of this variant.

Answer: We obtained the information that two cases with the Δ12+68 variant were epidemiologically closely related (new data in Table 1). These patients were part of one of the few transmission clusters in Uruguay during that period; we added this information in the results.

- "The topology of the phylogenetic tree implies that the Δ12 variant emerged early in the N.7 lineage during its initial differentiation in Uruguay (the first case was reported in July 2020); the Δ12+68 variant emerged subsequently by the acquisition of a second deletion (Δ68) from a Δ12 individual." I agree this is probably most likely, but any possibility of under-sequenced communities or provinces where an earlier emergence of the Δ68 might have been missed?

Answer: The emergence of Δ12 before Δ68 is supported by: i) the few cases occurring in the period or analysis, ii) the fact that Δ12 appeared two months earlier, and iii) the double-deleted samples were genetically homogenous and epidemiologically related (at least two of the four cases). Furthermore, considering that all N7 we analyzed have the Δ12 deletion, the later emergence of Δ68 seems more parsimonious. We agree with the reviewer that there is a chance of a different scenario, but our data does not support it. 

- I find the discussion on structure and function of the putative ORF7ab gene product to be very speculative (line 280 onwards). Should be edited and condensed heavily to avoid speculations (such as "the Δ12 mutation in ORF7a could be slightly deleterious and temporarily tolerated in a low-competitive population; after acquiring the additional deletion (Δ68)"). Instead, just state the type of experiments that could be performed to determine effects of deletions observed.

Answer: We reduced and moderated the tone of the discussion from line 280 onwards. We highlighted in yellow the paragraphs that were modified in the discussion section. Thanks for the suggestion; we believe the discussion is now clearer and concise. 

- Should add a section on limitations of the study.

Answer: We added a statement in the discussion commenting on the limited number of samples due to Uruguay's low prevalence that underpowers statistical analysis (i.e., frequency of cases). Our analysis, therefore, represents a snapshot in time before the emergence of variants of concern that replace the N.7 lineage. Another limitation was the method of sampling. Individuals submitted for the report may not fully reflect the prevalence of infections in the overall population. Nevertheless, the data obtained were from different geographic origins, and the sample number can be considered relatively representative. Finally, another potential limitation of the study may be the lack of functional studies to conclude the virus fitness. In Uruguay, we limited our analysis to genetic variability because we do not have facilities to culture the virus but recognize the need for these studies. Despite the above limitations, we believe that our research results contribute to understanding indels origin and prevalence by providing new insights into the genome variability and dynamics of accessory ORFs within SARS-CoV-2 lineages.

Editor comments

As pointed out by the reviewer, while the technical analysis and writing are done very well, there is an overabundance of speculative discussion in the manuscript. It would take much more data and analysis to untangle the combined effect of viral fitness, population genetics, and public health measures on the spread and prevalence of particular SARS-CoV-2 lineages in this pandemic. We appreciate that these hypotheses were presented sensibly, but politely request that the discussions from line 280 onwards be scaled back to strike a more balanced tone. Please also address the other points raised by the reviewer below.

Answer: We thank the editor for the positive comments. Following the suggestions, we reduced the most speculative part of the discussion and moderated some sentences' tone (highlighted in yellow). 

Answer: References were checked and modified accordingly. 

Additional changes

Typos and spelling mistakes were corrected throughout the manuscript. References suggested by the reviewers were included, and figures and tables were checked

---

## [Editor Report · Decision Letter 1]

24 Jan 2022

Consecutive deletions in a unique Uruguayan SARS-CoV-2 lineage evidence the genetic variability potential of accessory genes

PONE-D-21-32425R1

Dear Dr. Perez,

We’re pleased to inform you that your manuscript has been judged scientifically suitable for publication and will be formally accepted for publication once it meets all outstanding technical requirements.

Kind regards,

Herman Tse

Academic Editor

PLOS ONE
---

## [Editor Report · Acceptance letter]

8 Feb 2022

PONE-D-21-32425R1 

Consecutive deletions in a unique Uruguayan SARS-CoV-2 lineage evidence the genetic variability potential of accessory genes 

Dear Dr. Pérez:

I'm pleased to inform you that your manuscript has been deemed suitable for publication in PLOS ONE. Congratulations! Your manuscript is now with our production department. 

Kind regards, 

on behalf of

Dr. Herman Tse 

Academic Editor

PLOS ONE